# Vibrio cholerae O139 genomes provide a clue to why it may have failed to usher in the eighth cholera pandemic

Thandavarayan Ramamurthy[1,12], Agila Kumari Pragasam[2,12], Alyce Taylor-Brown[3,12], Robert C. Will [4,12], Karthick Vasudevan[2,5], Bhabatosh Das [6], Sunil Kumar Srivastava[7], Goutam Chowdhury [1], Asish K. Mukhopadhyay[1], Shanta Dutta[1], Balaji Veeraraghavan[2], Nicholas R. Thomson [3,8], Naresh C. Sharma[9], Gopinath Balakrish Nair[10], Yoshifumi Takeda[11], Amit Ghosh[1], Gordon Dougan[4] & Ankur Mutreja [4,6✉]

Cholera is a life-threatening infectious disease that remains an important public health issue in several low and middle-income countries. In 1992, a newly identified O139 *Vibrio cholerae* temporarily displaced the O1 serogroup. No study has been able to answer why the potential eighth cholera pandemic (8CP) causing *V. cholerae* O139 emerged so successfully and then died out. We conducted a genomic study, including 330 O139 isolates, covering emergence of the serogroup in 1992 through to 2015. We noted two key genomic evolutionary changes that may have been responsible for the disappearance of genetically distinct but temporally overlapping waves (A-C) of O139. Firstly, as the waves progressed, a switch from a homogenous toxin genotype in wave-A to heterogeneous genotypes. Secondly, a gradual loss of antimicrobial resistance (AMR) with the progression of waves. We hypothesize that these two changes contributed to the eventual epidemiological decline of O139.

[1] National Institute of Cholera and Enteric Diseases (NICED), Kolkata, West Bengal, India. [2] Department of Clinical Microbiology, Christian Medical College, Vellore, India. [3] Parasites & Microbes Programme, Wellcome Sanger Institute, Wellcome Genome Campus, Hinxton, Cambridge CB10 1SA, United Kingdom. [4] Cambridge Institute of Therapeutic Immunology & Infectious Disease (CITIID), Department of Medicine, University of Cambridge, Cambridge, United Kingdom. [5] School of Applied Sciences, Department of Biotechnology, REVA University, Bangalore, India. [6] Translational Health Science and Technology Institute, Faridabad, India. [7] Swami Shraddhanand College, University of Delhi, New Delhi, India. [8] London School of Hygiene and Tropical Medicine, WC1EHT London, United Kingdom. [9] Maharishi Valmiki Infectious Diseases Hospital, Delhi, India. [10] Rajiv Gandhi Centre for Biotechnology, Trivandrum, India. [11] National Institute of Infectious Diseases, Tokyo, Japan. [12]These authors contributed equally: Thandavarayan Ramamurthy, Agila Kumari Pragasam, Alyce Taylor-Brown, Robert C Will. ✉email: am872@medschl.cam.ac.uk

**V**ibrio cholerae is a Gram-negative, comma-shaped bacterium, which causes outbreaks of acute diarrheal disease, cholera, through ingestion of contaminated water and food. This bacterium inhabits aquatic environments and can cause cholera outbreaks in endemic areas and regions affected by natural or man-made disasters. *V. cholerae* is broadly classified into >200 serogroups based on the lipopolysaccharide (LPS) component (O antigen) of the cell wall, and out of these, only O1 and O139 are known to cause epidemics and pandemics[1]. In contrast, the non-O1/O139 *V. cholerae*, which also reside in the environment, generally cause sporadic diseases with mild 'cholera-like diarrhea' and food-associated outbreaks[2–5]. According to the World Health Organization (WHO), the global burden of cholera is estimated to be 1.3 to 4.0 million cases with 21,000 to 143,000 deaths every year[6]. Until now, there have been seven pandemics of cholera, all of which have been caused by O1 serogroup *V. cholerae*, with the first one documented in 1817. Of the two O1 biotypes, classical and El Tor, the classical biotype was the primary cause until the sixth pandemic but by 1923, it discontinued from the epidemiological scenario[7]. The on-going seventh cholera pandemic was caused by strains belonging to the seventh pandemic El Tor (7PET) lineage, which began with a large cholera outbreak in Indonesia in 1961 and then spread across South Asia in the next two years. Subsequently, the El Tor *V. cholerae* spread to Africa in the 1970s, South America in the 1990s and the Caribbean Islands in 2010[8–10]. The pandemic persists today with large outbreaks in Yemen and Somalia[11,12].

A unique non-O1 *V. cholerae* emerged in the early 1990s and caused disease that closely resembled that of *V. cholerae* O1 El Tor[13]. First identified in Madras (now known as Chennai), southern India, this strain did not agglutinate with any of the then-existing 138 somatic antisera. Thus, the strain was placed into a novel serogroup, later named O139. This serogroup bears the synonym "Bengal" due to its first appearance in the vicinity of the Bay of Bengal[13,14]. Generally, the non-O1 serogroups of *V. cholerae* do not produce cholera toxin (CT). In contrast, the O139 *V. cholerae* strains were found to carry the CT encoding genes (*ctxAB*), sometimes in multiple copies[15–17]. Whole-genome sequence-based phylogenetic analysis of 7PET strains showed the existence of three independent but overlapping transmission waves[8]. A phylogenetic tree placed strains of *V. cholerae* O139 within wave 2 of 7PET, residing in their own distinct 7PET sublineage[8]. *V. cholerae* O139 most likely emerged through the homologous recombination at the *gmhD* gene and IS*1358* that are present upstream and downstream of the *wbe* cluster conferring the O antigen phenotype[18]. This resulted in the replacement of a 22 kb *wbe* region of the O1 antigen with 35 kb *wbf* gene cluster encoding the O139 antigen[19]. Biosynthesis of the O139 antigen appeared to be similar to that in a strain of the O22 serogroup, which might have been the original donor[18].

Following its emergence, *V. cholerae* O139 temporarily dominated epidemiologically over the O1 El Tor biotype in several parts of India and Bangladesh, causing large cholera outbreaks in India and Southern Bangladesh, and continued to spread across Asia until the mid-2000s[20,21]. During the initial surge of cholera cases in 1992–93, *V. cholerae* O139 has largely displaced the El Tor vibrios. Due to its surprising epidemiological success, it was genuinely feared to have the potential to usher in the eighth pandemic[22,23]. The belief was so strong at the time that the cholera vaccines in development were modified to include coverage for the O139 serogroup[24].

*V. cholerae* O139 strains from its first epidemic in 1992–93 were believed to be a unique clone derived from an El Tor O1 strain. However, subsequent molecular studies revealed considerable genetic diversity within the O139 genomes including variable copy numbers of the CTX prophage[15–17]. The first

identification of an integrative conjugative element (ICE) carrying genes encoding resistance to sulfamethoxazole/trimethoprim (SXT) was in an O139 strain isolated in India during 1993[25]. Prior to this, O139 strains were resistant to SXT and streptomycin, but still susceptible to tetracycline. However, the O139 strains isolated in 1996 were susceptible to SXT, due to the deletion of a 3.6 kb region encoding resistance genes to the SXT and streptomycin[26]. Later studies showed an increasing trend of resistance to ampicillin and neomycin, while continuing susceptibility to chloramphenicol and streptomycin[15,26,27]. This rapid shift in the antimicrobial resistance pattern due to mobile genetic elements is similar to the phenomenon currently seen in *V. cholerae* O1 strains globally[28–31].

Although substantial epidemiological and molecular information is available for *V. cholerae* O139, the genomic factors behind its emergence, subsequent spread and surprising disappearance remain unexplored. More importantly, the questions around the failure of *V. cholerae* O139 to initiate the eighth cholera pandemic remain unanswered. While no *V. cholerae* O139 associated cholera cases have been reported since 2015, a key factor in the public health significance of this serogroup is its genetic 7PET backbone. For this reason, in this work we have conducted a genomic study on *V. cholerae* O139 by sequencing the strains collected at the time of its emergence in 1992, through to 2015. We aimed to investigate the potential of *V. cholerae* O139 to cause eighth cholera pandemic (8CP) and why it disappeared from the epidemiological landscape through genomic and evolutionary signatures.

## Results

**Serogroup O139 *V. cholerae* is more diverse than previously thought**. We assessed the phylogenetic positions of *V. cholerae* O139 genomes ($n = 330$) within the global species phylogeny (Fig. 1a). This revealed that the majority ($n = 321$) of the newly sequenced genomes clustered with a very limited number of previously published, publicly available *V. cholerae* O139 serogroup genomes, forming a distinct sublineage of 7PET wave-2 (Fig. 1b). Multilocus sequence type (MLST) derived from whole-genome sequence revealed that all the strains belong to same seventh pandemic clonal type ST69 (as do serogroup O1 genomes). Nine O139 *V. cholerae* genomes, however, were placed outside of this sublineage, interspersed throughout the broader *V. cholerae* species phylogeny (Fig. 1a).

Seven genomes were related to non-7PET lineages and strains. The most ancestral strain, NPO_566, from Nagpur, India isolated during 1994 more closely clustered with a Latin American serogroup O12 clinical isolate from 1994 (strain 1587; accession number AAUR01000000). Another strain, AS_9, isolated in Kolkata in 1995, clustered with strains that preceded the endemic Latin American 5 clade (ELA-5), while four strains from Kolkata isolated between 1993 and 2010 (CO_151, IDH_02927, CO_853b and CO_853a) resided between ELA-1 and ELA-2, related to non-O1 Haitian strains from the 2010 outbreak (Fig. 1a). Three of these strains, CO_853b, CO_853a and IDH_02927 harbored a typical O139 LPS operon (Supplementary Fig. 1). In addition, CO_151 carried an ICE SXT element and an array of antimicrobial resistance (AMR) encoding genes such as *dfrA18* (trimethoprim resistance), *sul2* (sulfonamide resistance), *floR* (florfenicol/chloramphenicol resistance), *strA* and *strB* (streptomycin resistance) within a composite transposon. Another isolate from Nagpur, (NPO_546, 1994) clustered with a non-O1 Haitian strain (AM_19226; accession number AATY01000000) and a Bangladeshi O39 isolate (HE25; accession number SRR210778). All seven non-7PET genomes were non-toxigenic and also lacked *tcpA* (encoding toxin co-regulated pilus), however they retained

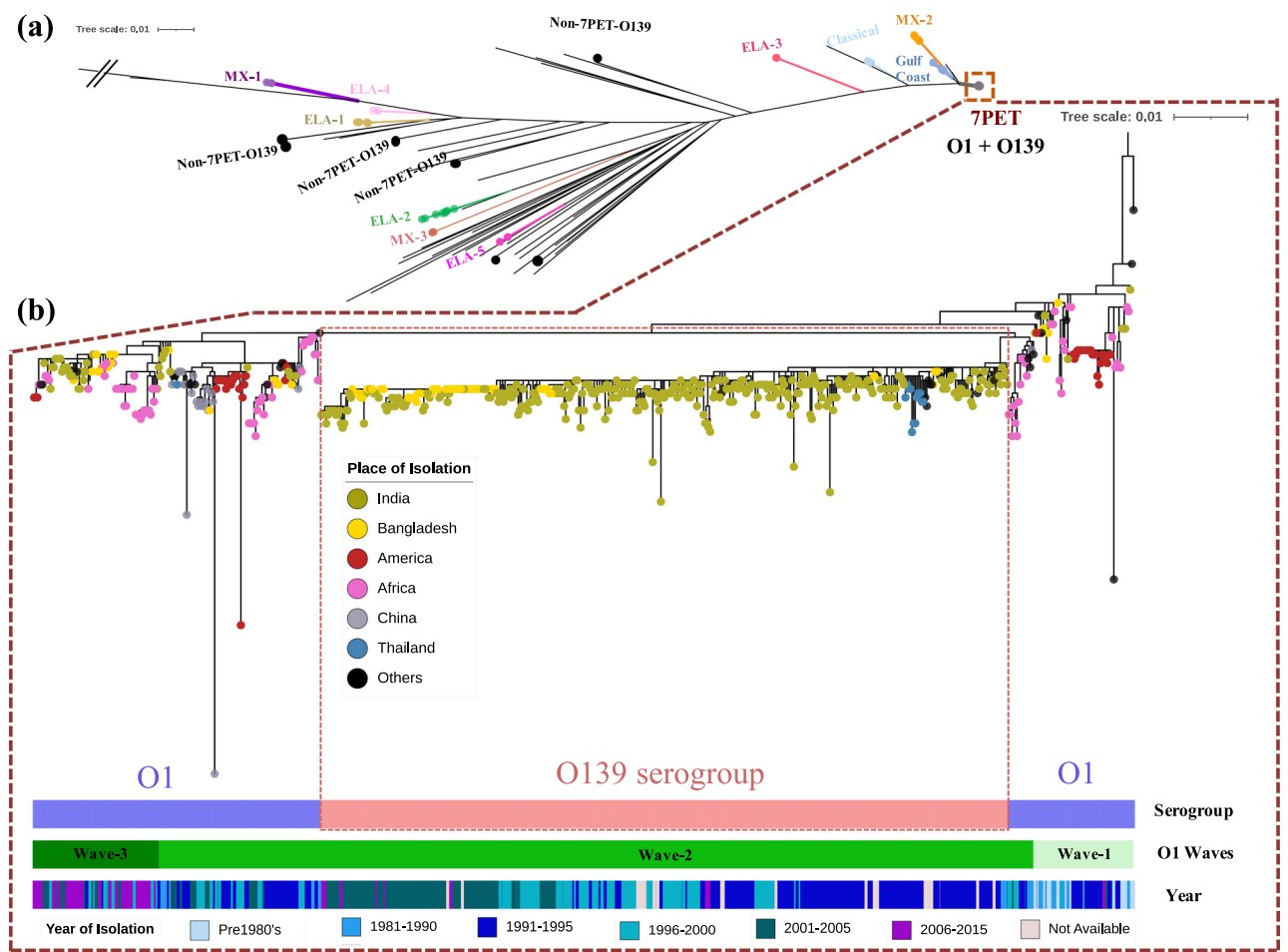

**Fig. 1 Phylogenetic distribution of V. cholerae O139. a** *V. cholerae* species phylogenetic tree constructed from a core gene SNP multiple sequence alignment. Colored branches represent *V. cholerae* serogroup O1 outbreak clades named as per previous publications, as labeled. Non-7PET *V. cholerae* O139 genomes are highlighted as black tree tips and the 7PET lineage is boxed. **b** Maximum likelihood phylogenetic tree of selected *V. cholerae* 7PET genomes. *V. cholerae* O139 ($n = 418$) genomes were placed into the context of the 7PET phylogeny using representatives from waves 1–3 ($n = 259$). The O139 sublineage is shown as a dashed box within the tree. Tree tips are colored by place of isolation (see key). Color bars depict the serogroup, transmission waves and year of isolation (see key). The tree was constructed from SNPs called against the *V. cholerae* N16961 reference genome and a pre-7PET isolate, M66, was used as an out-group to root the tree. Recombination events were removed using Gubbins. The scale bar represents the number of substitutions per site per genome.

different sets of virulence genes including the haemolysin and repeats in toxin encoding genes (*hylA*, *rtxA*, respectively) and the transcriptional activator *toxR*. Two possessed the mannose-sensitive haemagglutinin pilin protein encoding gene *mshA* as well (Supplementary Table 1). Four of the seven isolates (NPO_546, NPO_566, AS_9, IDH_02927) carried genes for a Type III secretion system, which has been commonly reported in non-toxigenic non-7PET strains associated with cholera-like diarrhea.

Within the 7PET lineage, two genomes clustered within a small distinct subclade of wave-2 of 7PET, distinct from the O139 sublineage. Both genomes, despite being phenotyped as O139, harbored an O antigen gene cluster that is characteristic of serogroup O1: ~28 Kb, lacking the *otnAB* and *wbeBF* genes, and possessing the O1-type *rfbV* and O1-specific *rfbG* gene and intact version of *rfbT* (Supplementary Fig. 1). These regions have high nucleotide identity with pre-7PET strains, which fit with the phylogenetic positions of these isolates. The first, S_36, isolated in Kolkata in 1993, shared its most recent common ancestor (MRCA) with THSTI_V12, an O1 Inaba isolate that also originated from Kolkata in 1989. The genetic composition of this isolate is interesting as it lacks the CTX phage including the

*ctxAB* subunits and the genes encoding the zonula occludens toxin (*zot*) and accessory cholera enterotoxin (ace), despite the presence of the gene *tcpA*, differentiating it from THSTI_V12, which has this region. The second isolate with this genotype, MDO_206, from Madurai in 1993, was found to be CTX positive (*ctxB_1*), and grouped with a Sri Lankan strain from 1981, as well as Mexican and Chinese isolates from the 1990s through to 2000. These genomes also carried the ICE SXT element encoding the AMR genes, *dfrA18, sul2, floR, strA* and *strB*, originally described in strains isolated during the global transmission of 7PET wave-2 during 1989–90. Also closely related is a strain that was serotyped as O36 but has an O1 genotype (accession number ERR163240), explaining often seen cross-reactivity of the somatic antigens with multiple antisera.

**The O139 locus in O139 serogroup of V. cholerae is consistent throughout the 7PET lineage.** Given the broad phylogenetic distribution of the O139 serogroup strains, we investigated the diversity of the O antigen gene cluster throughout our dataset. The majority of genomes ($n = 313$) harbored a "typical" O139 genotype, characterized by a region between approximately

42 Kbp in length (including the flanking genes *rfaD* which is involved in the synthesis of the LPS core and hypoosmotic stress protective mechanosensitive channel (*mscM*), encoding the O139-specific LPS type genes *wbfB*, *wbfE* and the capsular synthesis and transport genes *otnAB* and *gfcB* (Supplementary Fig. 1). The typical gene arrangement also consisted of an acidic amino acid triad, DDE (motif consists of two aspartic acid residues and a glutamic acid residue) domain transposase. This region is remarkably syntenic throughout this subset, with some exceptions for contig breaks, and some length variation (39,138–42,967 bp; average 42,855 bp) giving strong support for a single acquisition event by an O1 El Tor strain prior to clonal expansion of the sublineage. Further support for this is the lower GC content exhibited by this region. This operon on average has GC content 11.5% lower than the rest of the genome. Only two 7PET O139 genomes had a unique, atypical O-antigen genotype with a locus resembling *Vibrio* seventh pandemic island-I (VSP-1) inserted between a nucleotidyltransferase domain and *mscM* (AM_64; Supplementary Fig. 1).

Interestingly, we observed the typical gene O antigen arrangement in three of the seven non-7PET O139 genomes (CO_853a, CO_853b & IDH_02927). The remaining four non-7PET genomes (NPO_546, NPO_566, AS_9 & CO_151) had atypical O139 genotypes, several possessing some but not all the hallmarks of the typical O139 genotype, and others having structural similarities to operons from non-O1 serogroup genomes (Supplementary Fig. 1). While isolates AS_9 and NPO_566 both harbored *wbfB* and *otnB* but not in the typical order, CO_151 lacked the characteristic *wbf* genes but did possess *otnB*. The latter was the only one of these five operons to also carry transposase genes, similar to those found in 48853_F01, a previously published non-toxigenic non-7PET O139 *V. cholerae* sequenced with long-read technology. Isolate NPO_566 had a remarkably similar genetic structure to TMA_21, a previously published non-O1 strain from Brazil (Supplementary Fig. 1), suggesting that these strains may belong to novel serogroups with high similarity to O139 resulting in phenotypic agglutination.

## Temporal phylogeography of the *V. cholerae* O139 epidemics in Asia.

Cholera outbreaks caused by *V. cholerae* O139 were seen widely across India throughout the period when this serogroup was prevalent, with diverse focal points emerging at different times. We assessed the spatiotemporal distribution of these *V. cholerae* O139 outbreak strains ($n = 418$) in the context of previously well-studied representatives of the O1 7PET lineage ($n = 253$) that spread in three independent transmission waves (Fig. 2). All O139 serogroup genomes clustered as a monophyletic lineage sharing a common ancestor with a 1989 7PET-wave-2 strain, THSTI_V12 (*ctxb3* type), isolated from Kolkata. Using Bayesian analysis of population structure (BAPS) clustering analysis, we discerned three distinct clades/lineages of the O139 sublineage, which represented the successive emergence and introduction events in the form of temporally overlapping transmission waves of the O139 lineage (Fig. 2a). For naming consistency with the widely accepted 7PET-waves-1, 2 and 3, we termed these as O139-waves-A, B and C.

The first clade, O139-wave-A, is highly multifurcated, characterized by many polytomies and some distinct subclusters (Fig. 2a) with many temporally lineated branches in the phylogenetic tree, as identified by Bayesian logistic regression analysis that specifies its emergence in 1992 in Madras and Kolkata (Fig. 2b, e). O139-wave-A subsequently spread to other Indian cities, and was the only wave to reach other Asian countries (Fig. 2a, b, e). Madras strains from 1992 were most closely related to Madurai strains from 1993, suggesting regional transmission from Madras to Madurai. In 1994 and 1995, a phylogenetically related *V. cholerae* O139 from Madras and Madurai also appeared in the nearby city of Vellore. Internationally, close relatives of these Indian O139 strains were reported in Malaysia and China (Fig. 2b), which formed a small subclade with a Bangladesh isolate, within a larger subclade including Kolkata and Nagpur isolates of 1992 and 1993 (Fig. 2a). A separate clade closely related to Nagpur strains from 1993 comprised strains from Bangladesh (1993), Myanmar and Thailand (1994, 1995) (Fig. 2a). O139-wave-A strains were only observed from 1992–98 (Fig. 2e), but this clade gave rise to O139-wave-B (Fig. 2a).

The second wave, O139-wave-B, spanning 1994 to 2015, temporally overlapped with the O139-Wave-A (Fig. 2e). First dominant in Nagpur in 1994, it did not persist here (Fig. 2c, e). Instead, it appeared in Kolkata in 1996 and remained in circulation there until 2000. These Kolkata strains were likely introduced into Bangladesh in 2002, where they evolved as a separate subclade (Fig. 2a, c). Similarly, a subclade of O139-wave-B evolved in Vellore between 1997 and 2000 after the introduction, most likely initiated from Kolkata (Fig. 2c). Strains detected in Aurangabad were also very closely related to strains from Kolkata (Fig. 2a, c), as were strains that appeared in Delhi in 2001 and again in Vellore in 2003 and 2005 (Fig. 2c, e).

Different from O139-wave-A and O139-wave-B, which are characterized by a few internal nodes and long branches, O139-wave-C had a bifurcated substructure, with more basal nodes and shorter terminal nodes, representative of early introductions followed by local clonal expansions (Fig. 2a). This wave was observed in Kolkata, Delhi, and Bangladesh, where it re-emerged in 2013–14. Except the Kolkata clades, each phylogenetic expansion is likely to have been seeded through Vellore, where O139-wave-C was first observed in 2001 (Fig. 2a, d, e). This wave overlapped with O139-wave-B, with both clades observed concomitantly in Bangladesh in 2002 and Vellore in 2003. Neither wave-B nor wave-C of O139 was ever observed outside of India or Bangladesh (Fig. 2c–e).

## Timed evolution of *V. cholerae* O139 within the 7PET.

Bayesian evolutionary analysis of O139 predicted that the most recent common ancestor (MRCA) of *V. cholerae* 7PET O1 and O139-wave-A existed in 1984 [95% highest posterior density (HPD) = 1984–1986] (Supplementary Fig. 2). Similarly, the MRCA for O139-wave-B and O139-wave-C existed in 1990 (95% HPD = 1989–1990) (Supplementary Fig. 2).

The single nucleotide polymorphism (SNP) accumulation rate for the entire O139 population was found to be 3.5 SNPs per genome$^{-1}$ year$^{-1}$ (Supplementary Fig. 3), which is slightly higher than the 3.3 SNPs per genome$^{-1}$ year$^{-1}$ reported for the 7PET-O1 lineages[8]. The correlation value of 0.6 based on the date of isolation and root to tip distance for the O139 sublineage indicates a consistent evolutionary temporal signal. Substantial differences in the rates of evolution within the O139 population with respect to the three O139 waves were noted. The predicted SNP accumulation rates for O139-waves-A, B and C were 4.5, 1.28 and 0.9 SNPs genome$^{-1}$ year$^{-1}$, respectively (Fig. 3a, Supplementary Fig. 3). This contrasts with the stable evolutionary rates of the O1 lineages of 7PET with 2.2, 2.8 and 2.9 SNPs for waves-1, 2 and 3, respectively. Further, the proportion of SNPs due to recombination in O139 population was only 2%, indicating very little recombination within the epidemic O139 lineage compared to 7PET-O1 lineages that showed over 13% SNPs due to recombination[8].

## The decline of *V. cholerae* O139 is closely linked with the loss of AMR determinants.

In order to investigate possible genomic

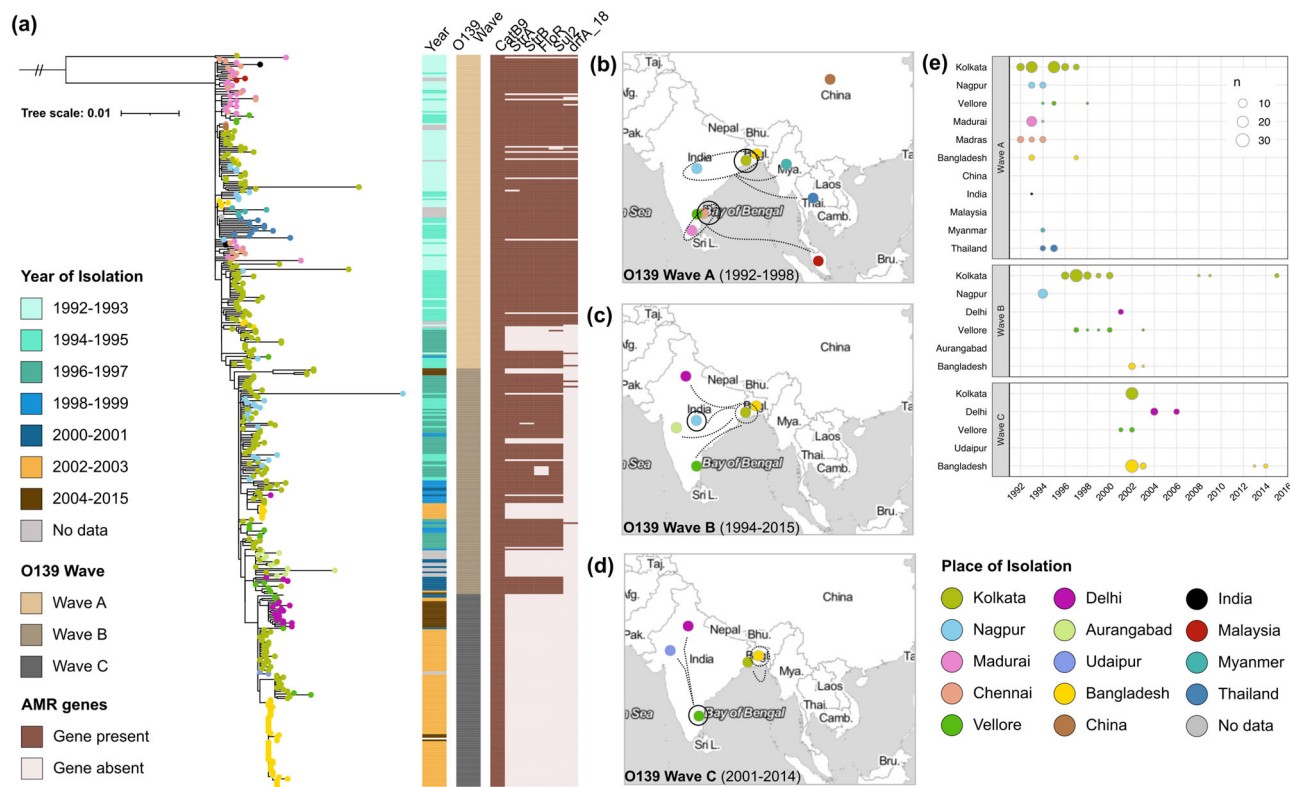

**Fig. 2 Phylogenetic and spatiotemporal dynamics of *V. cholerae* 7PET O139. a** Maximum likelihood phylogenetic tree of the *V. cholerae* 7PET O139 sublineage, constructed from whole-genome SNPs relative to the [H01] reference genome. Tree tips are colored by place of isolation (see key). Color strips and heat map depict the year of isolation, O139 wave, and AMR profile, respectively (see keys). The scale bar represents the number of substitutions per site per genome. **b–d** Spatiotemporal distribution of the three waves of *V. cholerae* 7PET O139 *V. cholerae* between 1992 and 2015. Solid circles represent the cities in which the clades were first observed, while dashed circles represent locations of secondary dominance. Dashed lines represent putative spread and circulation of the O139 waves. Map tiles were obtained from Stamen Design, under CC BY 3.0. Data by OpenStreetMap, under ODbL. **e** The number of genomes corresponding to each wave, over time. Points are colored by place of isolation and size of the point is relative to the number of genomes (see keys). Locations labeled on the Y axis with no corresponding X axis point are locations in which the O139 waves were observed, but for which the year of isolation is unknown and hence cannot be plotted. Where available, we have used higher resolution city information, but were restricted to country-level information for some of the samples.

explanations for the successive regression of each wave of the O139 lineage, we screened the genomes against a panel of AMR genes, pathogenicity islands and virulence genes[8]. We observed remarkable differences in AMR profiles between the different O139 waves: there was an evident decrease in the number of AMR genes as the O139 waves progressed in time (Figs. 1a and 3b). As the O139 lineage evolved, the same decreasing AMR trend was observed with the increase in phylogenetic root-to-tip (RTT) distance (Fig. 3c). These patterns are summarized per O139 wave in Fig. 3d.

During the emergence from their MRCA shared with a 7PET-O1 strain, O139-wave-A strains carried the ICE SXT element (also previously called as SXT[MO10]) (Fig. 2a), which contains a multidrug resistance (MDR) gene cassette with *floR*, *strAB*, *sul2* and *dfrA18*. The *floR*, *strAB* and *sul2* clustered together in a composite transposon like structure, with *dfrA18* positioned on a separate cassette. Although ICE SXT was carried by most of the O139-wave-A strains, 13% of isolates within O139-wave-A carried an ICE SXT core backbone without the MDR gene cassette, dispersed across the epidemic regions during 1992 and 1997 (Fig. 2a). However, towards the end of their dominant period (1996–97), strains of O139-wave-A did not possess *dfrA18* from the cassette, but continued to circulate until 1998. O139-wave-B strains initially carried *floR*, *strAB* and *sul2* genes on their ICE SXT (also previously called as ICE*Vch*Ind4), while only five strains harbored *dfrA18*, consistent with the AMR pattern that

appeared towards the epidemiological decline of O139-wave-A (Figs. 2a and 3b, d). However, unlike O139-wave-A, succeeding strains in this clade had no evidence of the MDR cassette (Fig. 2a). This genotype was also characteristic of O139-wave-C (Fig. 2a). Throughout the O139 epidemics, all strains continued to carry the chromosomally encoded *catB9* gene conferring chloramphenicol resistance (Fig. 2a).

Despite the differences in the presence of AMR genes in each of the O139 waves, the core ICE SXT backbone that carried the AMR gene cassette differed by just 36 SNPs in the entire O139 population studied, signifying a single point acquisition event. Similarly, the ICE SXT core of O139-wave-B (lacking *dfrA18*) differed only by three SNPs from that of the O139-wave-A (Supplementary Fig. 4). This is further supported by the fact that the loss of AMR in the O139 waves could be attributed to the presence of transposase genes flanking these AMR genes. This could be explained by one of the following hypotheses: a loss of *dfrA18* in O139-wave-A gave rise to O139-wave-B, followed by loss of other genes (*floR*, *strAB*, *sul2*) leading to O139-wave-C; or a complete loss of the entire AMR gene cassette from O139-wave-A leading to O139-wave-C. The latter, however, is more likely to have occurred, due to the epidemic period that was shared by O139-wave-A and C, as compared to that shared by O139-wave-A and B. Also, the occurrence of strains that lacked all the AMR genes in wave-A itself supports the latter (Fig. 2a). Further, upon comparing O139 with the epidemiologically overlapping 7PET

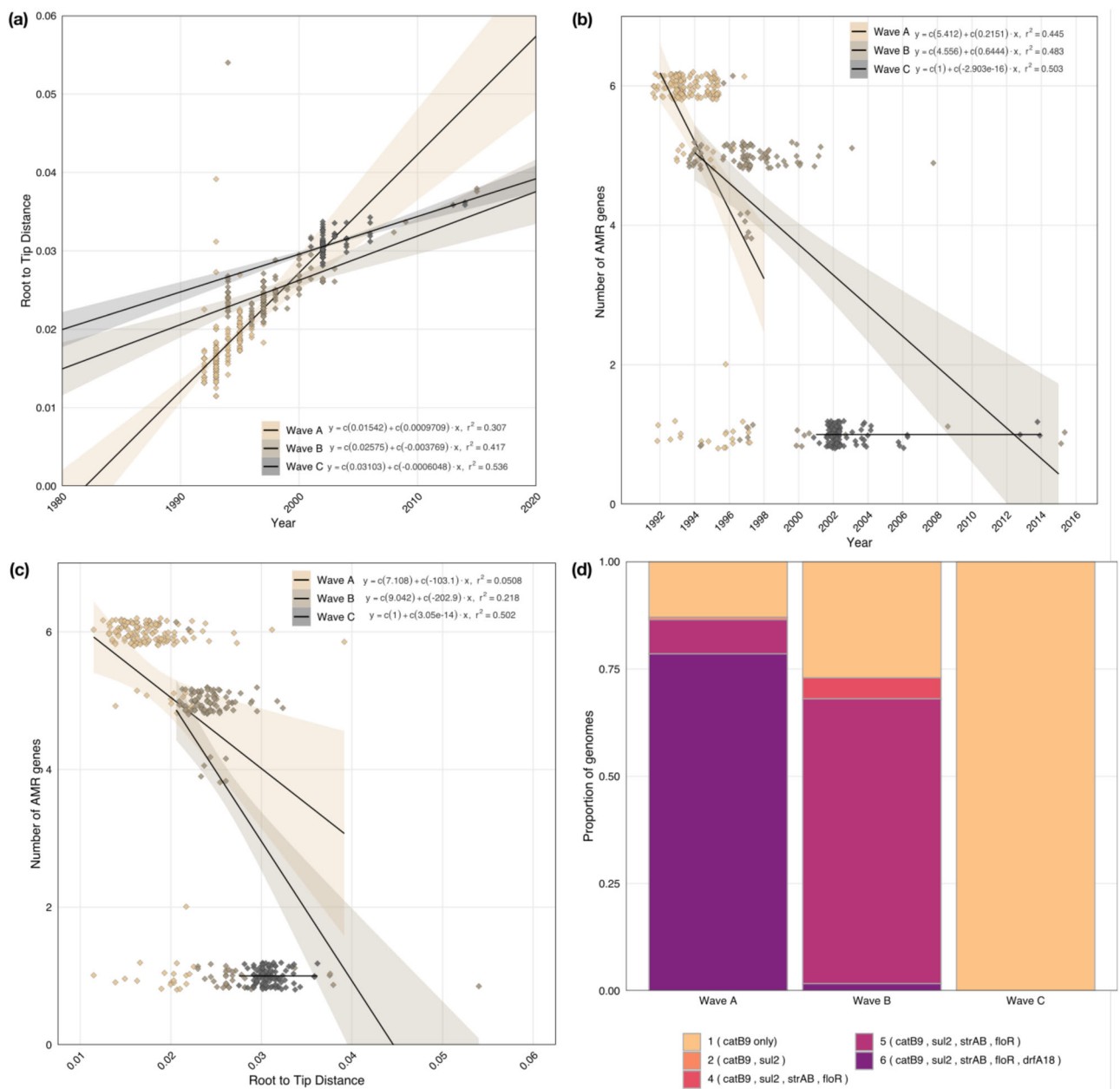

**Fig. 3 Evolutionary dynamics of *V. cholerae* O139. a–c** Linear regression plots of (**a**) the relationship between the RTT and year of isolation, i.e., the SNP accumulation rate of each of the O139 waves, (**b**) the number of AMR genes per genome, plotted per year of isolation, (**c**) the relationship between the number of AMR genes and root to tip distance (RTT). For (**a–c**), points are colored by wave (see key). The mean RTT is depicted by the linear regression line, with standard error depicted by the shaded bars. $R^2$ values and equations of the lines are also shown in the plots. **d** The proportion of varying numbers of AMR genes carried by each of the O139 waves/clades. Bars are colored by number of genes (see key).

O1 strains, large differences in the AMR profile were observed (Supplementary Fig. 5). In particular, 7PET wave-2 and wave-3 strains appeared to be multi drug resistant with the presence of acquired genes *(catB, dfrA, sul2, floR, strAB)*. Additionally, accumulation of double QRDR mutations in *gyrA* was exclusively observed in 7PET wave-3 strains, while single mutants were high in O139 wave-C as compared to others strains (Supplementary Fig. 5).

**The *V. cholerae* 7PET O139 waves possess other genetic signatures.** In addition to their AMR gene profiles, the *V. cholerae* 7PET O139 clades could also be differentiated by their *ctxB* type

(Supplementary Fig. 6). O139-wave-A strains carried the signature *ctxB3* El Tor variant, which matches the variant recorded in the wave-2 strains of 7PET-O1, while O139-wave-B and C strains carried *ctxB4* or *ctxB5*. We noted that ~9% ($n = 38$) of O139 serogroup strains from this study were non-toxigenic, and these were found in all the O139 waves A-C. Interestingly, all but two of the *ctxB*-negative isolates were *tcpA* positive and all O139 genomes, including 19/38 of the *ctx* negative strains, encoded the El Tor type *rstR* gene. Twenty-three strains showed the presence of classical *rstR* along with the El Tor *rstT* type. Notably, the *tcpA* gene of wave-A strains were 100% identical to that of El Tor variant of 7PET wave-2 strains, while wave-B and C differed from the El Tor variants by two SNPs. Also, there was a limited

variation in other virulence markers throughout the 7PET O139 lineage. Except for *rtxA*, which was missing from five O139-wave-B isolates, other key gene markers such as *als, makA, ompU, mshA, vasX, hlyA, vgrG* and *toxR* were present in almost all strains (99.5%, Supplementary Fig. 6). The two toxin genes *zot* and *ace*, encoded on the CTX phage, were missing from 29 of the *ctxB*-negative strains. Interestingly, in the clusters of O139-wave-C genomes that lacked the aforementioned AMR genes, *zot* and *ace* were retained, suggesting a complex, repeated emergence and decline of *V. cholerae* O139 in the Indian subcontinent. Despite these differences, analyses of the entire *ctx* phage region (~25 kb) in the study isolates revealed its presence in all the O139 genomes with a few isolates harboring partial regions when *ctxB* was absent (Supplementary Fig. 6).

Other than the *ctxB* variants, there were only minor differences in other genomic regions such as the *Vibrio* pathogenicity islands (VPI-1) and *Vibrio* seventh pandemic islands (VSP-1 and VSP-2), except for VPI-2. Screening of islands using a representative gene showed that VPI-2 and VSP-1 were ubiquitous in the O139 genomes, while VPI-1 was only missing from three genomes. VSP-2 was missing in 7% ($n = 27$) of the study isolates. The majority of these belonged to O139-wave-A while all O139-wave-C isolates contained this region (Supplementary Fig. 7). The presence of these islands was predicted based on an *in-silico* screening of a characteristic marker gene for each island (rather the entire islands), hence partial islands may be present in the assemblies in the absence of a marker gene. Further, on analyzing the presence of complete regions by mapping against the regions from N16961 genome, a few isolates have found to harbor partial islands of VPI-1, VSP-1 and VSP-2 (Supplementary Fig. 7). In VPI-2, all the O139 with the exception of two isolates lacked nearly one third of the island encoding type I restriction modification systems *nan-nag* region including the most important neuraminidase gene (*nanH*); whereas only the phage-like region has been retained (~20 kb).

The pan genome of *V. cholerae* O139 comprised a total of 6362 genes. While 3147 genes were present in 99–100% of the strains, representing the core genome, 239 genes formed the "soft core" genome (present in 95–99% of genomes) and 235 genes were shared by 15–95% of the strains, together representing intermediate frequency accessory genes. The presence of 2741 genes only in 1–15% of the strains (rare accessory genes; Supplementary Fig. 8a) suggests that a dynamic loss or gain gene flux played a huge role in the evolution of O139. However, comparing the gene frequencies between individual waves of O139, we did not find any O139 wave-specific core genes, or any core genes uniquely absent from any of the individual O139 subclades. We did not find any genes that were core to waves A or B and rare in wave C that weren't also present in O1 genomes (Supplementary Fig. 8b). Further, although we identified 11 genes core to O139-wave-C, these genes were found at intermediate frequencies across the O1 and O139 waves. Lastly, the lineage-specific genes we identified in this dataset were either at intermediate frequency (O1 waves), or rare (both O1 and O139 waves; Supplementary Fig. 8c) (Supplementary Table 2). These findings further demonstrate the complexity of the genome dynamics of the 7PET O139 sublineage.

## Discussion

The main aim of our study was to investigate why O139 *V. cholerae* subsided after its initial successful emergence and spread. Once envisaged to be the possible precursor of the eighth cholera pandemic, serogroup O139 was able to epidemiologically dominate O1 and establish itself as a major cause of cholera epidemics in India and Bangladesh. It was capable of successfully prevailing in the same geographical area[32], as well as in spreading to nearby

countries. Our study, analyzing the most complete collection of O139 *V. cholerae* available today, provides genomic insights into what may have led to the epidemiological and clinical disappearance of *V. cholerae* O139, and ultimately, its failure to seed the eighth cholera pandemic.

Previous studies showed that *V. cholerae* O139 is fully contained within its own monophyletic group, evolving from a single shared O1 El Tor ancestor sometime during the late 1980s, near the start of wave-2 of 7PET. Our study, consisting of 330 newly sequenced *V. cholerae* O139 genomes further refines earlier findings and reveals significant levels of genomic changes that may have eventually led to its unexpected decline.

Our analysis showed three genetically distinct but temporally overlapping clades of *V. cholerae* O139 that emerged in India and subsequently spread through South Asia. The O139 sublineage underwent rapid genetic change and diversification in wave-A before a slower diversification period during waves B and C. We observed two main genomic changes associated with key AMR and virulence that may, in combination or in different weights, might be responsible for the disappearance of the epidemic O139 lineage. Firstly, central to the transition between clades that co-existed in some of the cities where major O139 outbreaks were reported, was a characteristic temporal loss of AMR genes. Secondly, with the wave progression there was a stark switch from a homogenous *ctxB3* genotype in O139-wave-A to a heterogeneous presence of five different *ctxB* alleles: *ctxB4, ctxB5*, and three novel variants in O139-wave-A (*ctxb−3 + F69L*, $n = 1$) and O139-wave-C (H20, D24, A28, H34, T36, Y39, F46, K55, T68; $n = 2$). Our data provides robust scientific context to the previous speculation by Faruque et al., (2003) that genetic changes might be responsible for the decline of *V. cholerae* O139[21]. Since O139 VPI-2 lacks the notable regions of the *nan-nag* gene clusters, its persistence in the environment and sensitivity of host cells to the cytotoxin (*ctxb*) has been highly limited[33]. Further, the absence of the *nan-nag* gene cluster in O139 may not allow the strains to grow in the mucus rich gut environment with high sialic acids[34]. This supports the lack of competitive advantage of O139 strains being more pathogenic in the host colonization as compared to O1 7PET strains that harbors complete VPI-2 with functional *nan-nag* regions. These findings concur with the study that reported the decreased competitive index in colonization-competition assays of strains lacking VPI-2 as compared to O1 El Tor strains, signifying the importance of utilization ability of sialic acids in the host environment[35].

Antimicrobial resistance remains a game changer in the epidemiology of the infectious diseases landscape, so too for the cholera pandemic as we highlight in this study. Owing to the use of sulphamethoxazole/trimethoprim during the 1970s, (7PET wave-1), the antimicrobial pressure has subsequently attributed to the acquisition of SXT/R391 ICE encoding multiple antimicrobial resistance determinants including *dfrA* and *sul* (along with *strAB* & *floR*). This eventually may have led to the population shift from wave-1 to wave-2 and early wave-3 strains during the 1990s, which has been dated back to 1978–84, suggesting the SXT circulation 10 years prior to O139 appearance. With the progression of the seventh cholera pandemic, nearly all the wave-3 O1 strains were multi drug resistant and found to harbor *dfrA, sul, strAB* and *tet* genes. Alongside the cholera pandemic during the mid 1990s, the burden of MDR typhoid further complicated the infectious disease landscape with high use of fluoroquinolones. As a huge proportion were eliminated in active forms, environmental contamination rates were reported to be high. This pressure has further favored the accumulation of QRDR mutations in the environmental *V. cholerae* providing a fitness advantage over others[36]. On the holistic view of the 7PET population, the dynamics in the AMR profile between the epidemiologically

overlapping O1 and O139 appears to be a significant factor for O139 disappearance, which has not been contemplated until now. The occurrence of a sequential AMR loss in O139 strains was in contrast to the increasing drug resistant phenotypes in O1 7PET wave-2 and wave-3 strains that shared the epidemiological landscape alongside O139. This see-saw effect supports the hypothesis that the highly competitive drug resistant O1 population with *gyrA* double serine mutants has an additional fitness advantage for its persistence compared to the O139 population.

Further, it has been previously thought that there were multiple progenitors of *V. cholerae* O139[37]. Here we showcase that while O139 *V. cholerae* did indeed originate from multiple ancestors, the epidemic and toxigenic O139 lineage that was thought to lead to the eighth pandemic had a single origin from within the O1 7PET-wave-2. These harbored the 7PET-specific gene marker (VC2346), further confirming their origin. We also found that all the epidemic O139 strains had the *tcpA* gene, (an El Tor variant with a difference of 1-2 SNPs) establishing that this O139 lineage must have come from a single 7PET O1 progenitor population. Outside this lineage, there was a heterogeneous mix of distantly related genomes harboring a diverse array of O antigen operon clusters, suggesting that new serogroups can also be acquired by environmental O1 or non-O1 *V. cholerae*. This highlights that when genomic configurations are non-7PET and non-toxigenic, the likelihood of acquisition of clinically relevant serogroups in incorrect backbones resulting in epidemiologically successful lineages is minimal. Overall, our data also suggests that although 7PET O139 *V. cholerae* has largely disappeared from the cholera epidemic scenario, there exists a real possibility for O139 or a different serogroup lineage to emerge from the 7PET-O1 lineage, facilitated by minimal genomic acquisition events. Thus, monitoring the 7PET-O1 lineage not just for the toxin and AMR genes but also for its serogroup prevalence should continue to be the key to stay ahead of the pathogen in its evolutionary track. In addition, for cholera vaccines to continue to be effective, it is vital that serogroup replacement in the 7PET lineage and any clonal spread of these new variants is assessed carefully to plan appropriate vaccine modification and public health action for cholera control.

## Materials and methods

**Bacterial isolation and antimicrobial susceptibility testing.** Clinical isolates of *V. cholerae* O139 serogroup ($n = 330$) isolated between 1993 and 2015 across several parts of India ($n = 316$) and other Asian regions, Bangladesh ($n = 5$), Myanmar ($n = 5$), China ($n = 2$) and Malaysia ($n = 2$) were included in this study. Isolates from India were recovered from the archival facility at the National Institute of Cholera and Enteric Diseases (NICED, Kolkata), a reference repository laboratory in India. These isolates were predominantly from the Eastern (Kolkata; $n = 194$), followed by Southern (Madurai; $n = 25$, Vellore; $n = 18$ and Madras; $n = 17$), Western (Nagpur; $n = -30$, Aurangabad; $n = 11$ and Udaipur; $n = 2$) and Northern (Delhi; $n = 19$) parts of India. *V. cholerae* O139 isolated between 1992 and 2015 were included in this study. All the isolates were re-confirmed using standard biochemical and serological methods.

**DNA preparation and whole-genome sequencing.** All the confirmed isolates were grown in Luria-Bertani broth (Difco) overnight at 37 °C and total nucleic acid was extracted using a bacterial Genomic DNA extraction kit (Qiagen) as per manufacturer's instructions. DNA was then quantified using a Nanodrop spectrophotometer. Genomic libraries were prepared for each isolate tagged using unique indexing/barcodes, followed by a paired end 125 bp sequencing done on an Illumina Hi Seq platform at the Wellcome Sanger Institute (WSI, Hinxton UK). The reads were then segregated based on the index of each sample, before performing the down-stream analysis. All the sequences were deposited in the public database European Nucleotide Archive-ENA and accession numbers are listed in Supplementary Data 1[38–42].

**Dataset curation, genome assembly and annotation.** In addition to the 330 *V. cholerae* O139 sequenced in this study, 340 genomes from previously published literatures were included (Supplementary Data 1). This includes genomes of 253 globally and phylogenetically representative *V. cholerae* O1 and 87 *V. cholerae*

O139 isolates. Within *V. cholerae* O1, a representative of each lineage (WASA, LAT:1-3, Haiti and Nepal clades) of 7PET strains of wave 1, 2 and 3 were chosen for comparative analysis purpose. Raw reads of these genomes were retrieved from ENA database using their accession numbers. WSI pipelines were used for the detailed genome analysis. FASTQ read files were assembled and annotated using SPAdes v.3.11.0 and PROKKA v.1.5 respectively[43–45].

**Phylogenetic analysis.** Initially, for the phylogenetic analysis of 7PET *V. cholerae*, raw paired-end Illumina reads were mapped against the reference genome *V. cholerae* El Tor strain N16961 (Accession number AE003852 and AE003853) using SMALT v.0.7.6. Following mapping, multi fasta alignment and indels files were joined to create an alignment file of all the genomes. Recombination regions were identified, filtered and removed from the alignment using Gubbins v.1.4.10, yielding a final alignment of 5057 SNP sites. A maximum likelihood phylogenetic tree was generated using RAxML v.8.2.8 on the final SNP alignment with 100 bootstrap replicates. A similar approach was followed to generate the phylogenetic tree for only *V. cholerae* O139 genomes ($n = 417$), for which raw reads were mapped to the recently published high quality reference genome of *V. cholerae* O139 serogroup 48852_H01 (accession numbers LT992488-LT992489)[46]. This final alignment yielded 3250 SNP sites for O139 genomes. The tree was annotated with metadata using EMBL Interactive Tree of Life iTOL v.5.0. To determine subclades of 7PET *V. cholerae* O139, we performed a hierarchical Bayesian Analysis of Population Structure (BAPS) as implemented in fastBAPS [Tonkin-Hill et al.[47]] using the "baps" prior, conditioned on the O139-only phylogeny outlined above.

**Antimicrobial resistance, cytotoxin gene and pathogenicity island detection.** SRST2 was used to retrieve AMR and virulence markers present in all the genomes (Inouye et al.[48]). Briefly, raw reads are mapped against a database of AMR genes using SAMTOOLS and BOWTIE 2, and the best match is selected. Additionally, the assemblies were screened through the web-based server (https://cge.cbs.dtu.dk/services/CholeraeFinder/) to determine the presence of *V. cholerae* gene markers such as 7PET-specific gene (VC2346), CTX genotypes, ICE SXT, AMR and pathogenicity islands such as VPI-1, 2 and VSP-1, 2. We predicted the antigenic serogroup for the newly sequenced genomes in silico by screening the assemblies against an in-house database using Hamburger (https://github.com/djw533/hamburger). Briefly, a database was constructed containing flanking and marker genes for O139, O1, and selected non-O1/non-O139 operons. Gene sequences were individually aligned using Clustal Omega v 1.2.4 [Sievers et al.[49]], followed by HMM construction using HMMER v 3.2.1 [McClure et al.[50]]. The genetic organisation was then plotted in R using the ggplot2 [Wickham et al.[51]] and gggenes (https://cran.r-project.org/web/packages/gggenes/index.html) packages.

*Core genome analysis.* Whole-genome sequences were used to assign sequence types based on the multi locus sequence type (MLST) database (https://pubmlst.org/vcholerae). Core and pan genomes were created using ROARY (3.11.2), from annotated GFF3 files output from PROKKA[52]. The alignment output is a concatenated list of genes present in the core genome (if the gene is present in 99% of genomes by default). Roary involves conversion of coding sequences into proteins, clustering using CD-HIT after removing any partials and BLASTP comparison of each individual sequence against all other sequences. Finally, statistical grouping are compared with the CD-HIT grouping, to split homologous groups of paralogs into ortholog groups[53]. We further classified the pangenome into lineage-dependent categories based on gene frequency within and among 7PET O1 and O139 lineages using Twilight (https://github.com/ghoresh11/twilight) [Horesh et al.[54]], using the default thresholds.

**Linear regression, Bayesian temporal analysis, and spatial analysis.** We determined the relationship between root-to-tip distances and sampling dates using TempEst v.1.5.1 (https://tree.bio.ed.ac.uk). The Hasegawa, Kishino and Yano model (HKY) substitution with different demographic models (Bayesian skyline, exponential and constant) was investigated (Suchard et al.[55]). Markov chain Monte Carlo runs (MCMC) was run in 5 independent chains for 300 million generations. The Bayesian skyline model with uncorrelated lognormal relaxed clock was strongly preferred under Bayes factor testing. A burn in of 20% was discarded from each run and resulting log files were combined using LogCombiner 1.8.1 (Rambaut et al.[56]). The convergence of each run was manually evaluated by inspecting the chain traces. The tree data obtained from BEAST was summarized using the program TreeAnnotator v.1.8.2 and visualized using Figtree v.1.4.4 (Rambaut et al.[57]). Root to tip distances were plotted against year for each sample using the ggtree [Yu et al.[58]], ape [Paradis & Schliep[59]], dplyr [Wickham et al.[60]] and ggplot2 [Wickham et al.[51]] packages in R. Spatial distribution of samples was plotted using ggmap [Kahle & Wickham[61]], over maptiles downloaded from Stamen Design (www.maps.stamen.com/toner), under a Creative Commons Attribution (CC BY 3.0) license (https://creativecommons.org/licenses/by/3.0/) using latitude and longitude data corresponding to place of isolation.

**Reporting summary**. Further information on research design is available in the Nature Research Reporting Summary linked to this article.

## Data availability

Genomes sequences analyzed in this study are available under the project id PRJEB20897. Accession IDs of the genomes are mentioned in Supplementary Data 1 https://www.ebi.ac.uk/ena/browser/view/PRJEB20897?show=reads.

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

## Acknowledgements

We would like to acknowledge the funding agencies of the study. This study was supported in part by the Indian National Science Academy (INSA/SP/SS/2019), Indian Council of Medical Research (ref. no: AMR/TF/55/13ECDII dated 23/10/2013), and National Academy of Sciences (NASI), India. Sequencing was supported by the Wellcome Sanger Institute. G.D. and A.M. were funded by the NIHR Cambridge Biomedical Research Centre and NIHR AMR Research Capital Funding Scheme [NIHR200640]. A.T.B. and N.R.T. were supported by Wellcome funding to the Sanger Institute (#206194 and 108413/A/15/D). This research was funded in part by the Wellcome Trust (Grant numbers 206194 and 108413/A/15/D). For the purpose of open access, the authors have applied a CC-BY public copyright licence to any Author-Accepted Manuscript version arising from this submission. The views expressed are those of the author(s) and not necessarily those of the NIHR or the Department of Health and Social Care.

## Author contributions

Conceptualization: T.R. and A.M. Methodology: T.R., A.K.P., A.T.B., R.W., K.V., A.M. Investigation: T.R., A.K.P., A.T.B., R.C.W., K.V., A.M. Visualization: T.R., A.K.P., A.T.B., R.C.W., K.V., A.M. Funding acquisition: A.M., G.D. Project administration: A.M. Supervision: A.M., G.D., N.R.T. Writing – original draft: T.R., A.K.P., A.T.B., R.W., A.M. Writing – review & editing: T.R., A.K.P., A.T.B., R.W., K.V., B.D., S.K.S., G.C., A.K.M., S.D., B.V., N.R.T., N.C.S., G.B.N., Y.T., A.G., G.D., A.M.

## Competing interests

The authors declare no competing interests.

## Ethics approval

Our research complies with all relevant ethical regulations of the University of Cambridge, United Kingdom.
