## [Peer Review File · Nature Communications]

Vibrio cholerae O139 genomes provide a clue to why it may have failed to usher in the eighth cholera pandemicREVIEWER COMMENTS

Reviewer #1 (Remarks to the Author):

The authors present a fascinating and absorbing account of the emergence and evolution of the *Vibrio cholerae* O139 lineage. The paper is very nice to read and the data compelling and very well presented. I provide comments for the authors consideration below.

Major Comments:

My major point for discussion is that I do not believe this study shows what the authors suggest in the title and the discussion, namely the reasons why O139 disappeared. There is beautiful evolutionary genomic analysis but the gene gain and loss events and the loss of AMR genes are correlative to the fate of O139 with no evidence for this resulting in a loss of fitness compared to O1. Indeed to do this kind of work one would need to consider what happened in O1 as O139 faded to try and determine what drove the difference in fitness. I think the paper is great and stands as a great piece of work on O139, but it doesnt show why O139 disappeared.

Line 351: What about genes core to wave C but not present in A or B. Isnt it possible that acquisition of something may have weakened its fitness comparative to O1?

Minor comments:

Line 29: Change "till" to "until"

Line 33: If it is the only collection then it is by definition the largest

Line 48: Clarify "specific serogroup"

Line164: clean up two consecutive statements in parentheses.

Line 361: Wording needs toned down. Dont need to say "that could possibly be put together"

Reviewer #2 (Remarks to the Author):

The manuscript by Ramamurthy et al explores the genomes of ~300 newly sequenced *V. cholerae* O139 isolates to provide insight into cholera epidemiology. O139 first emerged in the early 1990's but did not displace the dominant O1 serogroup as the cause of pandemic cholera, despite a fitness advantage in evading serogroup specific immunity. The general pattern of newly emerging lineages and their subsequent rise to dominance or failure to supplant previously circulating *V. cholerae* lineages is well documented but poorly understood, thus the overall topic of this manuscript is highly significant. Following their analysis, the authors posit that O139 strains failed to dominate because of a lack of AMR genes. The authors could significantly strengthen their argument with a comparable analysis of O1 strains (perhaps also wave 2 strains); did O1 7PET-Wave-2 strains maintain more AMR genes over the same period, or perhaps did wave 3 strains displace the entire wave 2 lineage (including O139 strains) because Wave 3 strains harbor more AMR genes? Of course, we cannot know for certain with this analysis, but the authors may find that the number of AMR genes overtime for lineages that did persist is indeed not different than the pattern observed in O139 strains, in which case their hypothesis would not be supported. Such a finding would not negate interest in or significance of the present manuscript, but nonetheless comparable analysis of O1 strains should be performed given it is on hand and well within the scope of the expertise of the authors and the topic of the manuscript. Further comments to improve the clarity of the manuscript are provided below:

Major:

There are several contradictory statements (often without references) regarding the circulation of O139 strains, for example in the Abstract the authors state that O139 'largely displaced VC O1 until 2015', but then in the introduction ""After the initial surge in O139 cholera cases in 1992-93, *V. cholerae* O139 was reported infrequently, however, due to its surprising epidemiological success, it was genuinely feared to have the potential to usher in the eighth pandemic.22, 23"

The authors are not using consistent or standard nomenclature to describe SXT ICEs, for example

in the Results section 1st paragraph: CO_151 carried an intSXT element –Then later say: These genomes also carried the ICE_SXT/R391 element, later call it ICE-SXTMO10 element. I suggest the authors consider recent publications and use consistent/standard nomenclature, for example see Bioteau et al 2018 AEM <https://doi.org/10.1128/AEM.00485-18> and LeGault et al 2021 Science and <https://doi.org/10.1128/AEM.00485-18>. This is particularly relevant to the AMR argument, but also given that the first SXT ICE in *V. cholerae* was discovered in an O139 strain (one of the first ones sequenced, which serves as a common reference for O139 *V. cholerae*), it is critical that correct and consistent nomenclature is applied.

Minor: (please note, no line numbers were provided, so I added on the document I have and am providing them in sparsely below, without line numbers it is much more difficult to convey my comments)

“Among the genetic and genomic diversity...” what is the difference between genetic / genomic?

Last intro paragraph “O139 has undergone genetic assortment...” Please clarify what genetic assortment is referring to in the context of bacterial evolution

In Fig 1 the black dots are referring to the 9 O139 strains that were placed outside of the 7PET–Wave-2 sublineage where all other O139 cluster, this was difficult to gather from the figure and it would be much easier to simply label those on the figure for emphasis & clarity. Also for Figure 1:
-The color switching is difficult to interpret at times. Perhaps just have regions circled and labeled directly?
-The branching in 1A is difficult to see. A smaller, but more fanned-out figure might be better.

The results heading ‘The O antigen in O139 *V. cholerae* is consistent throughout the 7PET lineage’ Is misleading given that the O139 O antigen locus is NOT consistent throughout the seventh pandemic lineage (given the lineage is largely O1)

There is a statement made “Only two 7PET O139 genomes had a unique, atypical O-antigen genotype with a locus resembling VSP-1 inserted between a nucleotidyltransferase domain and mscM (AM_64; S. Figure 1).” –S Fig 1 – shows ONE 7PET atypical 23385_1#114 which looks like it has an insertion – how is this ‘VSP-1 like?’. Please explain.

The authors state they have an O139 phenotyped with typical O1 operon – can they validate this phenotype or was there a mixup in the strains?

There is a statement “The first clade, O139-Wave-A, is highly multifurcated, characterised by polytomies” -this terminology should be simplified / explained

Please clarify “13% of isolates within O139-Wave-A carried an ICE-SXTMO10 without the MDR gene cassette, dispersed across the epidemic regions during 1992 and 1997” -did they carry a different ice, or is it that they harbor the MO10 ICE with a deletion?

“However, unlike O139-Wave-A, later strains in this clade had no evidence of the MDR cassette” – again, did they have no ice, or different ICE with different MDR cassette?

The statement “the core intSXT backbone that carried the AMR gene cassette differed just by 36 SNPs in the entire O139 population studied, signifying a single point acquisition event” is not accurate. The entire SXT ICE with variable gene content needs to be evaluated given the modular nature of SXT ICEs, just looking at the integrase is not informative regarding acquisition of SXT ICEs in all strains.

The statement “The two toxin genes zot and ace, encoded on the CTX phage, were missing from 29 of the ctxB negative strains” is unclear, do these lack ctx entirely or n=38 are just toxin minus?

Figure 2:

-Mix of cities and countries in legends are confusing. The use of similar but slightly different colors in subpanels a & e is unfortunate.

- Are there isolates in panel A that are not present in Panel E? The absence of some of the locations in the legends would suggest so.
- Can the authors add more resolution on the time axis for panel E (i.e. more years labeled)
- Malaysia does not even appear on either location legend?
- Is the color for Chennai placed on China in panel B??
- If the authors are going to assert transmission from one location to another, then using arrows instead of dashed lines would be helpful in panels B,C,D

Lines 174-252 The inconsistencies in Fig 2 make a lot of the observations in this section hard to follow

Line 220 "it was the only wave to reach other Asian countries" the city/country color issues in figure 2 make this claim hard to verify.

Line 223 "First seen in Aurangabad in 1993" This strain does not appear to be in panel E

Line 257 "O139-Wave-B and O139-Wave-C" these waves are labeled as 1-3 in FigS2 legend

Line 260 "3.5 SNPs per genome-1 year⁻¹ (Figure 3a)" This does not appear to be the correct figure for this data??

Figure 3:

- There appears to be a mixup with the legend and which panel is which.
- For the AMR panel, perhaps it would be better to have individual columns for each AMR within each wave. As is it is hard to immediately understand.
- Is "root to tip" the same as "SNPs per genome per year"? If not, there is a serious discrepancy between the figure and the text.
- There is no R-squared value for the AMR panel, it should be included
- Figure 3b – why root to tip in X axis? Should be overtime instead?

Line 341 "hence partial islands may be present in the assemblies in the absence of a marker gene" The authors should check this.

Line 359 "re-emerging multiple times" It seems possible it was only occasionally observed due to limited sampling, rather than re-emerging. Figure 3c would suggest that it is pretty continuously present.

"Our study, examining the most comprehensive and only collection of O139 V. cholerae that could possibly be put together" This claim ('could possibly be put together') is excessive.

Line 379 "stark switch from a homogenous ctxB3 genotype" Can the authors reference any data about the effects of these CTX genotypes? Epidemiologically or in models?

Line 380 "five different ctxB genes" Please include the mutational profiles of these genotypes in a supplemental figure.

There is a slight discrepancy in the counts of genomes: "339 V. cholerae O139 sequenced in this study" It looks to me like 340 are labeled as "this study". And instead of "284 genomes from previously published literatures" there appears to be 330.

Supplementary table 1 – all references are not in the main references – what is Domman et al 2018? All strains need to have publicly available accession numbers; what is "Original_Rob" this appears to be a mix of accession numbers and internal numbers for strains/data that are not publicly accessible.

Query	Responses
Reviewer #1 (Remarks to the Author): The authors present a fascinating and absorbing account of the emergence and evolution of the Vibrio cholerae O139 lineage. The paper is very nice to read and the data compelling and very well presented. I provide comments for the authors consideration below.	
My major point for discussion is that I do not believe this study shows what the authors suggest in the title and the discussion, namely the reasons why O139 disappeared. There is beautiful evolutionary genomic analysis but the gene gain and loss events and the loss of AMR genes are correlative to the fate of O139 with no evidence for this resulting in a loss of fitness compared to O1. Indeed to do this kind of work one would need to consider what happened in O1 as O139 faded to try and determine what drove the difference in fitness. I think the paper is great and stands as a great piece of work on O139, but it doesn't show why O139 disappeared.	As suggested by this reviewer, we have modified the important points including the title of the manuscript. We have also analysed the V. cholerae O1 sequences during change in the trend of O139 serogroup to substantiate our findings and justify the original title as much as possible. A supplement figure has now been included (S. Figure 5) highlighting the differences in the highly drug resistant O1 waves Vs. sequential AMR loss in O139 waves. This has been detailed in the results and as well as discussion accordingly. However, since there may also be some phenotypic and regulatory changes involved, we have made a slight tweak to the title by adding "may have".
Line 351: What about genes core to wave C but not present in A or B. Isn't it possible that acquisition of something may have weakened its fitness comparative to O1?	We have re-examined the pangenome data and added further details to the manuscript (S. Figure 8). There were no genes core to any of the O139 waves that weren't present in any other group.

Minor comments:	We have corrected this in the revised manuscript
Line 29: Change "till" to "until"	
Line 33: If it is the only collection then it is by definition the largest	In the revised version, we have changed the content as follows “We conducted a comprehensive genomic study with the collection of V. cholerae O139 covering isolates from the time of its emergence in 1992 through to 2015”. We have made similar changes in the discussion section also.
Line 48: Clarify "specific serogroup"	We have removed “with specific serogroups” from this line, as the serogroups and their disease associations are detailed later in the paragraph.
Line 361: Wording needs toned down. Dont need to say "that could possibly be put together"	This statement now reads “most comprehensive collection of O139 V. cholerae to date,”
Line164: clean up two consecutive statements in parentheses.	The first set of parentheses were unnecessary and have been removed.
Reviewer #2 (Remarks to the Author): The manuscript by Ramamurthy et al explores the genomes of ~300 newly sequenced V. cholerae O139 isolates to provide insight into cholera epidemiology. O139 first emerged in the early 1990’s but did not displace the dominant O1 serogroup as the cause of pandemic cholera, despite a fitness advantage in evading serogroup specific immunity. The general pattern of newly emerging lineages and their subsequent rise to dominance or failure to supplant previously circulating V. cholerae lineages is well documented but poorly understood, thus the overall topic of this manuscript is highly significant. Following their analysis, the authors posit that O139 strains failed to dominate because of a lack of AMR genes. The authors could significantly strengthen their argument with a comparable	

analysis of O1 strains (perhaps also wave 2 strains); did O1 7PET-Wave-2 strains maintain more AMR genes over the same period, or perhaps did wave 3 strains displace the entire wave 2 lineage (including O139 strains) because Wave 3 strains harbor more AMR genes? Of course, we cannot know for certain with this analysis, but the authors may find that the number of AMR genes overtime for lineages that did persist is indeed not different than the pattern observed in O139 strains, in which case their hypothesis would not be supported. Such a finding would not negate interest in or significance of the present manuscript, but nonetheless comparable analysis of O1 strains should be performed given it is on hand and well within the scope of the expertise of the authors and the topic of the manuscript. Further comments to improve the clarity of the manuscript are provided below:

Major:

There are several contradictory statements (often without references) regarding the circulation of O139 strains, for example in the Abstract the authors state that O139 ‘largely displaced VC O1 until 2015’, but then in the introduction ““After the initial surge in O139 cholera cases in 1992-93, *V. cholerae* O139 was reported infrequently, however, due to its surprising epidemiological success, it was genuinely feared to have the potential to usher in the eighth pandemic.22, 23”

Thank you for highlighting. We have corrected and simplified the statement in the introduction to “After the initial surge in O139 cholera cases in 1992-93, *V. cholerae* O139 was genuinely feared to have the potential to usher in the eighth pandemic.22, 23”. We have also replaced 2015 with 2002 in the abstract.

The authors are not using consistent or standard nomenclature to describe SXT ICEs, for example in the Results section 1st paragraph: CO_151 carried an intSXT element –Then later say: These genomes also carried the ICE_SXT/R391 element, later call it ICE-SXTMO10 element. I suggest the authors consider recent publications and use consistent/standard nomenclature, for example see Bioteau et al 2018 AEM https://doi.org/10.1128/AEM.00485-18 and LeGault et al 2021 Science and https://doi.org/10.1128/AEM.00485-18. This is particularly relevant to the AMR argument, but also given that the first SXT ICE in V. cholerae was discovered in an O139 strain (one of the first ones sequenced, which serves as a common reference for O139 V. cholerae), it is critical that correct and consistent nomenclature is applied.	We have now made the text consistent by using “ICE SXT” as the terminology throughout the manuscript for two main reasons that are: (1) S. Figure 4 on the core ICE SXT phylogenetic view was derived based on the O139 SXT element as a reference AY055428. (2) The differences in the AMR carried by ICE SXT were termed due to the presence of dfr18, strAB, sul2 and floR as “int-SXT-MO10”, while ICE SXT carrying strAB, sul2 and floR were termed as ICEVchInd4 (Wozniak et al., 2009). However, both belonged to ICE SXT family and hence we have made the terminology now uniform by using ICE SXT.
Minor: (please note, no line numbers were provided, so I added on the document I have and am providing them in sparsely below, without line numbers it is much more difficult to convey my comments) “Among the genetic and genomic diversity...” what is the difference between genetic / genomic?	We have corrected this to read “among the genomic diversity”
Last intro paragraph “O139 has undergone genetic assortment...” Please clarify what genetic assortment is referring to in the context of bacterial evolution	This has now been removed
In Fig 1 the black dots are referring to the 9 O139 strains that	Figure 1 A and B have now been revised as per the suggested

were placed outside of the 7PET–Wave-2 sublineage where all other O139 cluster, this was difficult to gather from the figure and it would be much easier to simply label those on the figure for emphasis & clarity. Also for Figure 1:  -The color switching is difficult to interpret at times. Perhaps just have regions circled and labeled directly? -The branching in 1A is difficult to see. A smaller, but more fanned-out figure might be better. 	changes.
The results heading “The O antigen in O139 V. cholerae is consistent throughout the 7PET lineage’ Is misleading given that the O139 O antigen locus is NOT consistent throughout the seventh pandemic lineage (given the lineage is largely O1)	We have corrected this heading to “The O139 locus V. cholerae is consistent throughout the 7PET lineage”
There is a statement made “Only two 7PET O139 genomes had a unique, atypical O-antigen genotype with a locus resembling VSP-1 inserted between a nucleotidyltransferase domain and mscM (AM_64; S. Figure 1).” –S Fig 1 – shows ONE 7PET atypical 23385_1#114 which looks like it has an insertion – how is this ‘VSP-1 like?’. Please explain.	Only one of the two genomes containing this region were selected for the figure. The gene module that has been inserted resembles that of VSP-I in its genetic content and arrangement. VSP-1 region has been found to be intact.
The authors state they have an O139 phenotyped with typical O1 operon – can they validate this phenotype or was there a mixup in the strains?	The isolates used in this study were all included because they were phenotyped as O139. We know anecdotally that some serogroups can cross-react, however we cannot prove this either way as we do not have sera for all known serogroups to test this, nor do we have genomic representatives for all known serogroups, let alone other novel serogroups that may exist, which we cannot rule out either. The phylogenetic placement of the two samples carrying this phenotype/genotype combination is interesting (distinct from the

	O139 sublineage and not in the same cluster as each other) and provides possible evidence for some transition period between wave 2 and the emergence of O139, although with only two genomes we reserved our speculation on this.
There is a statement “The first clade, O139-Wave-A, is highly multifurcated, characterised by polytomies” -this terminology should be simplified / explained	For clarity, the sentence has been simplified as “The first clade, O139-wave-A, is characterised by polytomies with many temporally lineated branches in the phylogenetic tree, as identified by Bayesian logistic regression analysis that specifies a sign of high genetic diversity”.
Please clarify “13% of isolates within O139-Wave-A carried an ICE-SXTMO10 without the MDR gene cassette, dispersed across the epidemic regions during 1992 and 1997” -did they carry a different ice, or is it that they harbor the MO10 ICE with a deletion? “However, unlike O139-Wave-A, later strains in this clade had no evidence of the MDR cassette” – again, did they have no ice, or different ICE with different MDR cassette?	As written in Line no: 311 – 324, all the Vibrio cholerae O139 genomes in this study harboured ICE SXT (as shown in S Figure 3) with a few SNP differences in it. The sentence has now been revised as “13% of isolates within O139-Wave-A carried an ICE SXT core backbone without the MDR gene cassette...”
The statement “the core intSXT backbone that carried the AMR gene cassette differed just by 36 SNPs in the entire O139 population studied, signifying a single point acquisition event” is not accurate. The entire SXT ICE with variable gene content needs to be evaluated given the modular nature of SXT ICEs, just looking at the integrase is not informative regarding acquisition of SXT ICEs in all strains.	We agree with the Reviewer’s concern. We did not make this statement just based on the presence of a single “integrase” gene of ICE SXT. Please note that we have included S. Figure 3 - Maximum likelihood phylogenetic tree of the core ICE SXT. All the Vibrio cholerae O139 genomes in this study were mapped against the entire ICE SXT core backbone (AY055428 - 51,961 kb region) to confirm its presence in the entire collection. The same backbone was

	used for analysing the genetic similarity, which showed only 36 SNPs in total among all the O139 genomes.
The statement “The two toxin genes zot and ace, encoded on the CTX phage, were missing from 29 of the ctxB negative strains” is unclear, do these lack ctx entirely or n=38 are just toxin minus?	We have analysed the presence of the entire CTX phage elements in all the study genomes. We found that all the genomes harboured CTX Phage, but only certain toxin genes such as ctxB/zot/ace were absent in a few isolates. A supplement figure has been added now as S. Figure 6 .
Figure 2:  -Mix of cities and countries in legends are confusing. The use of similar but slightly different colors in subpanels a & e is unfortunate. -Are there isolates in panel A that are not present in Panel E? The absence of some of the locations in the legends would suggest so. -Can the authors add more resolution on the time axis for panel E (i.e. more years labeled) -Malaysia does not even appear on either location legend? -Is the color for Chennai placed on China in panel B?? -If the authors are going to assert transmission from one location to another, then using arrows instead of dashed lines would be helpful in panels B,C,D. 	 - We were restricted to country-level information for some of the samples, whereas chose to use the higher resolution city information where it was available. We agree this is unfortunate but comes with the territory of opportunistic sampling and retrospective sequencing. We have noted this in the figure legend. - Yes - we do not have the location information or year of collection for some of the samples. - We have extended the year labels as suggested. - we have clarified the colour legends for figure 2 to now include Malaysia [and others that were accidentally omitted]. - This was an oversight and this figure has now been corrected. - We are not strictly trying to assert transmission, as it is difficult to determine directionality. Additionally, we are basing this analysis of genomes, rather than cases, so have clarified this in the text and figure legend.
Lines 174-252 The inconsistencies in Fig 2 make a lot of the observations in this section hard to follow	We have made extensive changes to Figure 2 and clarified several parts of the text with the aim of making this section easier to follow.

Line 220 "it was the only wave to reach other Asian countries" the city/country color issues in figure 2 make this claim hard to verify.	This should be clearer now with the changes to Figure 2 outlined above.
Line 223 "First seen in Aurangabad in 1993" This strain does not appear to be in panel E	Thank you for spotting this. This has been corrected in the revised figure and text.
Line 257 "O139-Wave-B and O139-Wave-C" these waves are labeled as 1-3 in FigS2 legend	This has been corrected in the revised manuscript.
Line 260 "3.5 SNPs per genome-1 year -1 (Figure 3a)" This does not appear to be the correct figure for this data??	We double-checked and found it to be the same.
Figure 3: -There appears to be a mixup with the legend and which panel is which. -For the AMR panel, perhaps it would be better to have individual columns for each AMR within each wave. As is it is hard to immediately understand. -Is "root to tip" the same as "SNPs per genome per year"? If not, there is a serious discrepancy between the figure and the text. -There is no R-squared value for the AMR panel, it should be included -Figure 3b – why root to tip in X axis? Should be overtime instead?	 - Yes, there was a mix up with the legend panels - this was a simple but poor oversight and has now been corrected. - We chose to have a stacked bar chart to emphasise the change in diversity/proportion of AMR genes carried by O139 strains over the successive O139 waves. - We have added the slope and correlation coefficient to these plots to clarify the relationships between SNPs and time. - R-squared value has now been included in the revised figure
Line 341 "hence partial islands may be present in the assemblies in the absence of a marker gene" The authors should	To check this, we have mapped all the study isolates against the entire regions of VPI-1 (VC_0817 to VC_0847: 40883 bp) VPI-2

check this.	(VC_1758 t VC_1809: 56770 bp), VSP-1 (VC_017 to VC_0186: 16679 bp) and VSP-2 (VC_0489 to VC_0517: 31,651 bp). We observed that the data presented in S. Figure 7 (presence depicted by a marker gene) was found to be concordant with the results of mapping entire regions of VPI-1 and VSP-2. Further, the mapped regions of entire islands have now been added to S. Figure 7.
Line 359 "re-emerging multiple times" It seems possible it was only occasionally observed due to limited sampling, rather than re-emerging. Figure 3c would suggest that it is pretty continuously present.	In the Discussion section, we have changed the wording to "Further, it was capable of successfully prevailed in the same geographical area, as well as in spreading to nearby countries".
"Our study, examining the most comprehensive and only collection of O139 V. cholerae that could possibly be put together" This claim ('could possibly be put together') is excessive.	This sentence was changed in the revised manuscript as "Our study, with the most comprehensive collection of O139 V. cholerae provides genomic insights into what led to the epidemiological and clinical disappearance of O139 V. cholerae, and ultimately, its failure to seed the eighth cholera pandemic".
Line 379 "stark switch from a homogenous ctxB3 genotype" Can the authors reference any data about the effects of these CTX genotypes? Epidemiologically or in models?	Studies have observed temporal fluctuation of ctxB genotypes in V. cholerae, suggesting a possible role of CTX phage in natural selection and short-term evolution (Bhuiyan et al., 2009; Rashid et al., 2016).
Line 380 "five different ctxB genes" Please include the mutational profiles of these genotypes in a supplemental figure.	This has been updated in the S. Figure 6 with the novel variants. Further, the substitutions are mentioned within the text. As these are a few, we preferred to mention this within the text, rather than a figure as suggested.

There is a slight discrepancy in the counts of genomes: "339 V. cholerae O139 sequenced in this study" It looks to me like 340 are labeled as "this study". And instead of "284 genomes from previously published literatures" there appears to be 330.	The discrepancy has been checked and updated accordingly. Clinical isolates of V. cholerae O139 serogroup (n=340) isolated between 1993 and 2015 across several parts of India (n=326) and other Asian regions, Bangladesh (n=5), Myanmar (n=5), China (n=2) and Malaysia (n=2) were included in this study. Isolates from India were recovered from the archival facility at the National Institute of Cholera and Enteric Diseases (NICED, Kolkata), a reference repository laboratory in India. These isolates were predominantly from the Eastern (Kolkata-194, followed by Southern (Madurai-25, Vellore-28 and Madras-17), Western (Nagpur-30, Aurangabad-11 and Udaipur-2) and Northern (Delhi-19) parts of India
Supplementary table 1 – all references are not in the main references – what is Domman et al 2018? All strains need to have publicly available accession numbers; what is "Original_Rob" this appears to be a mix of accession numbers and internal numbers for strains/data that are not publicly accessible	Missed out references from the Supplementary data has been now added in the revised paper and publicly available accession numbers of the study isolates were also updated in place of internal IDs.

REVIEWERS' COMMENTS

Reviewer #1 (Remarks to the Author):

In the interests of fairness I have reviewed this revised version of the manuscript purely in the context of my own original reviewer suggestions.

I thank the authors for taking my suggestions on board, especially around the claim of identifying the reason why O139 disappeared. I think the alterations do not hinder the quality of the paper at all whilst portraying a more realistic interpretation of the data

Reviewer #2 (Remarks to the Author):

Reviewer 1: . Indeed to do this kind of work one would need to consider what happened in O1 as O139 faded to try and determine what drove the difference in fitness.

My comment in previous review (which was not explicitly responded to) "The authors could significantly strengthen their argument with a comparable analysis of O1 strains (perhaps also wave 2 strains); did O1 7PET-Wave-2 strains maintain more AMR genes over the same period, or perhaps did wave 3 strains displace the entire wave 2 lineage (including O139 strains) because Wave 3 strains harbor more AMR genes?"

Given that both reviewers see this as a critical point, the analysis in Fig S5 should be moved to the main text, on it the authors should delineate the O139 waves (A,B,C).

As before, I do not agree with the statement "the core ICE SXT core backbone that carried the AMR gene cassette differed by just 36 SNPs in the entire O139 population studied, signifying a single point acquisition event." it is inappropriate to consider only the conserved core backbone of the SXT ICEs - they are modular, by definition their backbones are conserved but variable cargo define the SXT ICE.

Line 106; suggest spelling out sulphamethoxazole/trimethoprim when referring to the antibiotic resistance profile or it can easily be mis-interpreted to say that the strains were 'susceptible to SXT' meaning acquisition of the element. same with line 430.

REVIEWERS' COMMENTS

Queries	Response
Reviewer #1 (Remarks to the Author): In the interests of fairness I have reviewed this revised version of the manuscript purely in the context of my own original reviewer suggestions. I thank the authors for taking my suggestions on board, especially around the claim of identifying the reason why O139 disappeared. I think the alterations do not hinder the quality of the paper at all whilst portraying a more realistic interpretation of the data	We thank the reviewer again for their input.
Reviewer #2 (Remarks to the Author): Reviewer 1: . Indeed to do this kind of work one would need to consider what happened in O1 as O139 faded to try and determine what drove the difference in fitness. My comment in previous review (which was not explicitly responded to) "The authors could significantly strengthen their argument with a comparable analysis of O1 strains (perhaps also wave 2 strains); did O1	As suggested by the reviewer, we have addressed the queries related to O139 comparison with O1 profile. To this, we included a new S. Figure 5 and its detailed explanation in the results and discussion section as below: Results: “Further, upon comparing O139 with the epidemiologically overlapping 7PET O1 strains, large differences in the AMR portfolio has been observed (S. Figure 5). In particular, 7PET wave-2 and wave-3 strains were appeared to be multi drug resistant with the presence of acquired genes (catB, dfrA, sul2, floR, strA & strB). Additionally, accumulation of double QRDR mutations in gyrA was exclusively observed in 7PET wave-3 strains, while single mutants were high in O139 wave-C as compared to others strains (S. Figure 5).”

7PET–Wave-2 strains maintain more AMR genes over the same period, or perhaps did wave 3 strains displace the entire wave 2 lineage (including O139 strains) because Wave 3 strains harbor more AMR genes?" Given that both reviewers see this as a critical point, the analysis in Fig S5 should be moved to the main text, on it the authors should delineate the O139 waves (A,B,C).	Discussion: “Antimicrobial resistance remains a game changer in the epidemiology of infectious diseases landscape, as so to the cholera pandemic as we highlight in this study. Owing to the use of SXT during 1970’s 7PET wave-1, the antimicrobial pressure has subsequently attributed to the acquisition of SXT/R391 ICE encoding multiple antimicrobial resistance determinants including dfrA and sul (along with catB9, strAB & floR). This eventually led to the population shift from wave-1 to wave-2 and early wave-3 strains during 1990's, which has been dated back to 1978-84, suggesting the SXT circulation 10 years prior to O139 appearance. With the progression of the seventh cholera pandemic, nearly all the wave-3 O1 strains were multi drug resistant and found to harbour dfrA, sul, catB9, strAB and tet genes. Alongside cholera pandemic during mid 1990’s, the burden of MDR typhoid further complicated the infectious disease landscape with high use of fluoroquinolones. As a huge proportion were eliminated in active forms, environmental contamination rates were reported to be high. This pressure has further favoured the accumulation of QRDR mutations in the environmental V. cholerae providing fitness advantage over others. 36 On the holistic panoramic view of 7PET population, the dynamics in the AMR portfolio between the epidemiologically overlapping O1 and O139 appears to be a significant factor for O139 disappearance, which has not been contemplated till date. The occurrence of a sequential AMR loss in O139 strains was in contrast to the increasing drug resistant phenotypes in O1 7PET wave-2 and wave-3 strains that shared epidemiological landscape alongside O139. This see-saw effect supports the highly competitive drug resistant O1 population with double serine mutants providing additional fitness advantage for its persistence till date, as the O139 population gave way.” As per the latest suggestion, S. Figure 5 has now been updated with the three waves (A,B,C) of O139 lineages delineated in the figure, similar to O1 waves
As before, I do not agree with the statement "the core ICE SXT core backbone that carried the AMR gene cassette differed by just 36 SNPs in the entire O139 population studied, signifying a single point acquisition event." it is inappropriate to consider only the conserved core backbone of the SXT	In the SXT^{MO10}, the AMR genes are located in a ~17.2kbp composite transposon like element. The variability in the ICE elements arises majorly with the AMR gene composition that are flanked by the transposon like structures. The nomenclature system of SXT ICE were termed based on the AMR genes. For instance, VchInd4 that carried all the AMR genes, except dfr18. However, it differed from SXT^{MO10} just by 13 SNPs (Wozniak et al., 2009, PLOS GENETICS)

ICEs - they are modular, by definition their backbones are conserved but variable cargo define the SXT ICE.	Outside the AMR genes, the core backbone of ICE shared 97% similarity. And for these reasons, we have considered the core backbone to understand the ICE SXT acquisition event in the O139 population and for a robust phylogenetic analysis of the ICE SXT evolution, we prefer conserved core backbone analysis.
Line 106; suggest spelling out sulphamethoxazole/trimethoprim when referring to the antibiotic resistance profile or it can easily be mis-interpreted to say that the strains were 'susceptible to SXT' meaning acquisition of the element. same with line 430.	Changes have been made as suggested.